# Research on the Spatial and Temporal Distribution Evolution and Sustainable Development Mechanism of Smart Health and Elderly Care Demonstration Bases Based on GIS

Xiaolong Chen [1,2], Bowen Chen [1,*], Hongfeng Zhang [1] and Cora Un In Wong [1,*]

1   Faculty of Humanities and Social Sciences, Macao Polytechnic University, Macao 999078, China; osakacool@hait.edu.cn (X.C.); hfengzhang@mpu.edu.mo (H.Z.)
2   Department of Management, Henan Institute of Technology, Xinxiang 453002, China
*   Correspondence: p2312561@mpu.edu.mo (B.C.); corawong@mpu.edu.mo (C.U.I.W.); Tel.: +86-17765960396 (B.C.); +853-8599-3263 (C.U.I.W)

**Abstract:** Utilizing GIS technology and spatial analysis methodologies, this study endeavours to delve into and grasp the localized attributes of the regional units under investigation from a geographical vantage point, as well as the interrelationships among these units. This endeavour encompasses the identification and quantification of developmental patterns, the assessment of trends, and the resolution of any intricate issues about geographical location to make prognostications and informed decisions. Classic spatial analysis techniques such as the geographic concentration index, kernel density analysis, Thiessen polygons, and spatial autocorrelation analysis (Moran's I index) are employed in this inquiry. Initially, the study utilized the nearest neighbour index and geographic concentration index to gauge the equilibrium, proximity, and concentration of the spatiotemporal distribution of smart health elderly care demonstration bases across 31 provinces in China. Upon confirming the spatial clustering and imbalance of the distribution of elderly care demonstration bases in China, kernel density analysis was applied to compute the density of point features surrounding each output raster cell and to visually represent the spatiotemporal distribution status of the bases. Finally, Thiessen polygons and spatial autocorrelation analysis (Moran's I index) were introduced to further elucidate and validate the spatial distribution patterns of the elderly care demonstration bases. The findings of the research reveal that smart health and elderly care bases in China manifest spatial clustering, predominantly concentrated in the central and eastern regions of the country. The overarching pattern embodies a spatial model characterized by a "concentration in three poles with multiple cores surrounding". Ultimately, the study offers recommendations for the nexus between three principal mechanisms: market-driven development mechanisms, policy-driven development mechanisms, and technology-driven development mechanisms, advocating for the further progression of intelligent construction to attain the sustainable development of demonstration bases. This research furnishes a scientific foundation for the planning and industrial advancement of pertinent departments.

**Keywords:** geographic information system; spatial evolution; smart elderly care; agglomeration; hotspot distribution





## 1. Introduction

In the traditional fabric of Chinese society, families were originally the primary caretakers for elderly individuals. However, as social industrialization and urbanization have deepened, there has been a widening gulf between urban and rural areas in terms of living standards and income levels. This has resulted in a marked increase in population mobility. In pursuit of better prospects and living conditions, an escalating number of middle-aged and young people are opting to depart from their ancestral lands, leading to a separation between parents and children and a rapid rise in "empty nest" households. Consequently,

the proportion of elderly individuals cohabiting with their offspring has dwindled. The projected elderly dependency ratio in China is expected to reach 28.92% by 2022, surpassing the child dependency ratio, with a continuing divergence between the two [1]. By the midpoint of the 21st century, it is estimated that the elderly dependency ratio in China will be close to 38 percentage points higher than the child dependency ratio, signifying that every two young individuals will need to support one elderly person [2].

The increasing physical and emotional distance between progeny and their progenitors, coupled with the frenetic pace of contemporary life and limited leisure time, have substantially elevated the challenges faced by the younger generation in caring for their elders. Balancing work, personal life, and eldercare has led to a significant surge in the financial costs associated with assuming the responsibility of caring for the elderly among the middle-aged and young populace. This has also diminished the feasibility of providing long-term unpaid care for the elderly and weakened the capacity of the younger generation to support the elderly [3]. Particularly in terms of day-to-day and medical care, there is a growing need for communal elderly care, and the demand and willingness of the elderly for such care are expanding and strengthening.

According to the latest data released by the National Bureau of Statistics of China, it took China only 21 years to transition into a deeply ageing society, a considerably shorter duration than the 64 years for the United States, 46 years for the United Kingdom, and 40 years for Germany [4]. As of the end of 2022, the population aged 60 and above in China stood at 280.04 million, constituting 19.8% of the total population, with 209.78 million aged 65 and above accounting for 14.9% of the total population [5]. China's life expectancy has risen from 43.7 years in 1960 to 77.93 years in 2022 [6,7]. In 2022, China's birth rate was 6.77 per 1000, the death rate was 7.37 per 1000, and the natural population growth rate was −0.60%, marking the first negative growth in total population and natural population growth in 61 years [8]. China has emerged as the country with the largest elderly population worldwide, and the pace of population ageing is significantly outstripping the global average [9]. According to the population projection results of the United Nations World Population Prospects 2022, the proportion of people aged 60 and above in China will see an average annual increase of 2.35% between 2015 and 2055. Before 2035, the proportion of the population aged 60–69 in the total elderly population in China will continue to exceed 50%. After 2035, the proportion of the oldest elderly population will grow rapidly, surpassing 50% of the elderly population. By 2050, the number of people aged 60 and above in China will surpass 500 million, constituting 38.81% of the total population. In sum, the data demonstrate that China's population structure has entered a highly aged stage after progressing through the phases of adultization, ageing formation, and accelerated ageing, and the situation of population ageing is exceedingly severe [9,10].

To a large extent, population ageing has evolved into a pivotal trend in China's social development, with other economic and social issues related to ageing assuming greater prominence. With the rapid progression of the ageing process, the burden of elderly care on society as a whole is growing increasingly onerous. Traditional familial care for the elderly is no longer sufficient to meet the demands of development, and there exists a robust call for and inclination toward communal elderly care.

The 19th National Congress of the Communist Party of China, in its solemn proclamation, envisioned a future where China intricately weave a tapestry of policies and a societal ambience that venerates the elderly, extols filial piety, and reveres those in the advanced stages of life [11]. The culmination of this vision outlook came to fruition in April of 2019 when the venerable General Office of the State Council promulgated the Opinions on Advancing the Maturation of Elderly Care Services (Guobanfa [2019] No. 5). Across the intricate expanse of local governance, legislative measures were deftly interwoven, all with the commendable aim of enhancing the quality and efficacy of services tailored for the elderly, heralding a new era of unparalleled elderly care. This paradigm shift became an indomitable necessity [12,13].

To realize the grand stratagem encapsulated in the Healthy China 2030 Planning Outline, a consortium comprising twenty-one ministries and commissions, directed by the venerable National Development and Reform Commission, harmonized a synchronized symphony of purpose. On the vernal equinox of 29 September 2019, they unveiled the Action Blueprint for the Fostering of the High-Quality Health Sector (2019–2022) (Development and Reform Society (Development and Reform Society) 2019). This masterful symphony aims to precipitate the evolution of the health sector, fostering the emergence of a health industry system brimming with profound significance and an eminently rational framework.

On 16 February 2017, China's Ministry of Industry and Information Technology, Ministry of Civil Affairs, and National Health and Family Planning Commission released the Smart Healthy Elderly Care Industry Development Action Plan (2017–2020) [14]. According to the Notice of the General Office of the Ministry of Industry and Information Technology, the General Office of the Ministry of Civil Affairs and the General Office of the National Health Commission on Carrying out the Selection of Pilot Demonstrations of Smart Healthy Elderly Care Applications in 2021 (Industry and Information Technology Department E-Letter [2021] No. 266) and Industrial and the Notice of the General Office of the Ministry of Information Technology on Carrying out the Selection of Pilot Demonstrations of Smart Healthy and Elderly Care Applications in 2021 (Industry and Information Department E-Letter [2021] No. 268), 49 units including the Xicheng District in Beijing were identified as the first batch of smart homes in the country, alongside the health care application pilot demonstration unit [15,16].

Population aging is an important trend in social development. Currently, China's elderly care model mainly presents three elderly care models: home care, community care, and institutional care [17]. With the burgeoning demand for elderly care, the enhancement of nursing home infrastructure in certain regions has seen continuous refinement in recent years. However, as a general trend, the more diminutive nursing homes persist as the predominant establishments. They aim to establish and improve the dynamic monitoring mechanism of elderly health by integrating high-quality medical resources and social elderly care service resources, promote the in-depth application of information technology and intelligent hardware in elderly care services, and improve the quality of the elderly through the Smart Healthy Elderly Care Application Demonstration Base, achieving a sense of gain and happiness [18,19]. Such efforts hold profound significance in laying the foundation for a sustainable and high-caliber support system tailored to the needs of the elderly.

## 2. Overview of the Study Area

This study takes China as the target research area, based on the geographical point spatial distribution model, and takes 89 smart health care demonstration bases announced by government departments five times as the research object (Figure 1). ArcGIS 10.2 software is used to analyze and compare the vector raster data, explore the spatiotemporal evolution of the smart health elderly care demonstration base, and conduct qualitative and quantitative analysis of geospatial visualization. It will realize the spatial optimization of the layout of the national smart health elderly care demonstration base, and then provide a theoretical basis for the healthy and sustainable development of the national smart health elderly care demonstration base.

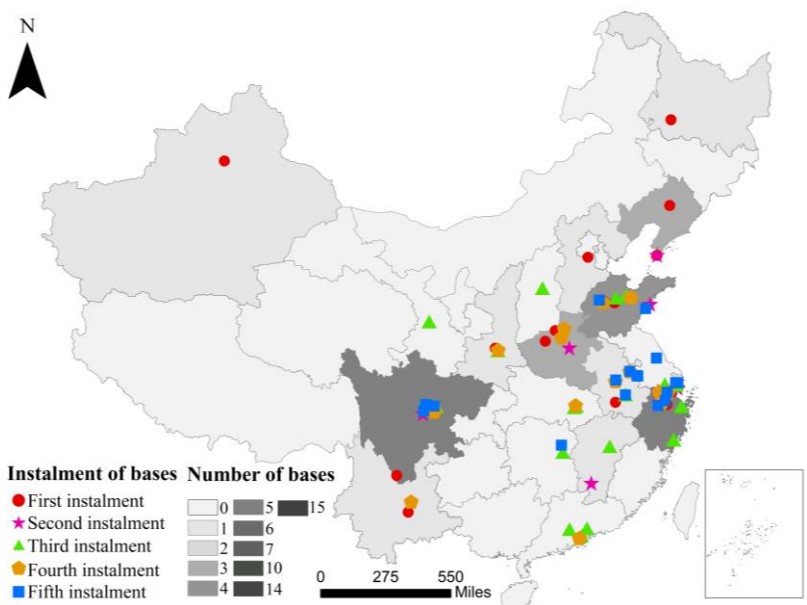

**Figure 1.** Overview map of the research region.

## 3. Materials and Methods

### 3.1. Research Methods

3.1.1. Nearest Neighbor Index Method

The nearest neighbor index is used to calculate the spatial distribution type of the smart health care demonstration base. The nearest neighbor index is often used to express the proximity of point-like things to each other in geographical space, represented by $R$. This geographical indicator is mainly used to analyze the spatial distribution of point-like things [20]. The calculation formula is as follows:

$$R_0 = \frac{1}{\sqrt[2]{\frac{n}{A}}} = \frac{1}{\sqrt[2]{D}} \tag{1}$$

$$R = R_0 / R_e \tag{2}$$

In Formula (1), $R_e$ is the theoretical nearest neighbor distance, $A$ is the area of the research area, and n is the number of smart healthy elderly care demonstration bases; in Formula (2), $R_o$ is the average nearest neighbor distance, and the nearest neighbor index ($R$). It is equal to the ratio of the actual average distance ($R_o$) and the theoretical nearest neighbor distance ($R_e$). When $R < 1$, it means that the spatial distribution of the smart health and elderly care demonstration base is clustered. The smaller the value, the more aggregated the spatial distribution of the smart and healthy elderly care demonstration base is. When $R = 1$, it means that the spatial distribution of the smart and healthy and elderly care demonstration base is random; when $R > 1$, it indicates that the spatial distribution of the smart healthy elderly care demonstration base is uniform. The larger the value, the more uniform the spatial distribution of the smart healthy elderly care demonstration base. Distribution characteristics of the research sample smart health elderly care demonstration bases were compared year by year, and according to the time series, 2001, 2007, 2012 and 2018, where the data had obvious changes, were used as interval time research targets. The sample data were rasterized, and geographical visualization tools were used to further analyze the evolution representation of the spatial pattern of the smart health and senior care demonstration base, and explore the spatiotemporal evolution rules of the smart health and senior care demonstration base.

### 3.1.2. Coefficient of Variation and PZ Value

The coefficient of variation, also known as the standard deviation rate, is another statistic that measures the degree of variation of each observation in the data [21,22]. When comparing the degree of variation of two or more data, if the unit of measurement is the same as the mean, the standard deviation can be used directly for comparison. If the units and/or averages are different, the standard deviation cannot be used to compare the degree of variation, but the ratio (relative value) of the standard deviation to the average must be used for comparison.

The ratio of the standard deviation to the mean is called the coefficient of variation, recorded as *CV* (coefficient of variance). The coefficient of variation can eliminate the influence of different units and/or averages on the comparison of the degree of variation of two or more data.

$$CV = \sigma/\mu \tag{3}$$

It mirrors the extent of dispersal concerning the unit mean and is frequently employed for juxtaposing the dispersion levels between two population means characterized by distinct values. If the means of two populations are equal, then comparing the standard deviation coefficients and comparing the standard deviations are equivalent. It represents the standard deviation variance and represents the average.

The *p*-value (*p*-value, probability, Pr) represents probability. It reflects the probability of an event occurring. In the analysis of spatial correlation, the *p*-value represents the probability that the observed spatial pattern was created by some random process. The official description of the Z score is also called the standard score and it is a process of dividing the difference between the score and the mean by the standard deviation.

### 3.1.3. Geographic Concentration Index

In order to explore the spatial distribution balance of the smart health elderly care demonstration base, it is necessary to study the concentration of its point element distribution. The geographical concentration index is an important indicator used to indicate the geographical concentration of a certain research object [23,24]. The formula is:

$$G = 100 \times \sqrt{\sum_{i=1}^{n} \left(\frac{x_i}{m}\right)^2} \tag{4}$$

Among them, *G* represents the geographical concentration index, which represents the number of *i*-th smart health care demonstration bases, *m* represents the total number of smart health care demonstration bases, and n represents the total number of regional administrative provinces. The *G* value of the geographical concentration index ranges from 0 to 100. The closer the *G* value is to the extreme point of 100, the more concentrated the smart health and senior care demonstration bases are. The closer the *G* value is to the extreme point of 0, the more dispersed the distribution of smart health and senior care demonstration bases will be.

### 3.1.4. Kernel Density Analysis

Kernel density analysis is used to calculate the density of point features around each output raster cell. For point elements, the density of points is estimated by the density of points in the surrounding neighborhood, which can indicate the aggregation status of point elements in the entire study area [25]. The calculation formula is:

$$R_n(X) = \frac{1}{n} \sum_{i=1}^{n} K\left(\frac{X - X_i}{h}\right) \tag{5}$$

Among them, $R_n(X)$ is the estimated value of kernel density occurring at a certain point X; h is the bandwidth; K is the kernel function; $(X - X_i)$ is the distance value from the estimated value point $X$ to the measured point $X_i$.

### 3.1.5. Thiessen Polygon

Thiessen polygon can use the coefficient of variation $C_V$ of the polygon area to estimate the degree of change in the area of convex polygons [26,27], thereby studying the distribution type of the sample. The calculation formula is:

$$r = \sqrt{\sum_{i=1}^{n} (S_i - S)^2 / n} \tag{6}$$

$$C_V = \frac{S}{r} \tag{7}$$

Among them, $S_i$ is the area of the $i$-th polygon; $S$ is the mean value of the polygon area; r is the standard deviation. According to Duyckaerts' research conclusion, when the $C_V$ is between 33% and 64%, it means that the sample points are randomly distributed; when the $C_V$ is greater than 64%, it means that the sample points are clustered; when the $C_V$ is less than 33%, it means that the sample points are uniformly distributed.

### 3.1.6. Spatial Autocorrelation Analysis

- Moran index

Moran index $I$ is an important indicator to measure spatial correlation, and is generally used to express the overall trend of spatial correlation of various variables in the entire study area [28,29]. The calculation formula is:

$$I = n \frac{\sum_{i=1}^{n} \sum_{j=1}^{n} w_i (x_i - \overline{x})(x_j - \overline{x})}{\sum_{i=1}^{n} \sum_{j=1}^{n} w_{ij} \sum_{i=1}^{n} (x_i - \overline{x})^2} \tag{8}$$

Among them, $x_i$ and $x_j$ are the mean number of sample points in area $i$ and $j$; $\omega_{ij}$ is the space vector matrix; $n$ is the total number of samples. The Moran index I is distributed in the interval $[-1, 1]$. When $I > 0$, it indicates that the sample points have a positive spatial correlation. The larger its value is, the more significant the spatial correlation is. When $I < 0$, it indicates that the sample points have a positive spatial correlation. For spatial negative correlation, the smaller the value, the greater the spatial difference; when $I = 0$, it indicates that the sample points are spatially random.

- Getis-Ord index

Moran index I can measure spatial correlation, but it cannot fully reflect the specific characteristics of correlation. The Getis-Ord index $G$ is generally used to analyze the degree of aggregation of local spatial observations, and can also observe the spatial distribution of hot and cold spots [30,31]. The calculation formula is:

$$G = \frac{\sum_{i=1}^{n} \sum_{j=1}^{n} w_{ij}(d) x_i x_j}{\sum_{i=1}^{n} \sum_{j=1}^{n} x_{ij}} (i \neq j) \tag{9}$$

When the result is positive, it means that the values around area $i$ are relatively high, which is a high-value spatial agglomeration; when the result is negative, it means that the values around area $i$ are relatively low, which is a low-value spatial aggregation.

### *3.2. Data Sources*

The data used in this article come from the list of smart health and elderly care demonstration bases published by the Ministry of Industry and Information Technology. The spatial data come from the map database of the National Basic Geographic Information Center (www.ngcc.cn) (accessed 1 October 2023). The spatial attributes such as the longitude and latitude of the smart health and elderly care demonstration base are obtained

with the help of the Baidu coordinate picker and Amap, and the spatial analysis is drawn using ArcGIS 10.2. Isometric graphics. The GDP of each province comes from the China Statistical Yearbook, as well as the provincial statistical yearbook and the provincial National Economic and Social Development Statistical Bulletin of the corresponding year. In view of the reliability of the data, the statistics in 2022 shall prevail.

## 4. Results

### 4.1. Spatial Distribution Types

ArcGIS 10.2 software was harnessed to apply the methodology of natural fracture classification in concert with Thiessen polygons for the cartographic depiction of the landscape adorned with China's smart healthcare facilities. The graphical manifestation of these endeavors is unveiled in Figure 2, an exquisite rendering of our findings. Within this visual tableau, the depth of hue in a given geographical region within the figure correlates with the sheer abundance of healthcare bases residing therein. Likewise, the intricacy of polygonal borders mirrors the compactness of the territory they demarcate, thereby elucidating the density of healthcare centers found within.

Evidently, the smart health and geriatric care facilities in China exhibit a discernible spatial agglomeration. These agglomerative enclaves are predominantly situated in the central and eastern domains of the nation, with three provinces, namely Zhejiang, Sichuan, and Shandong, standing out as conspicuous exemplars of this geographic clustering.

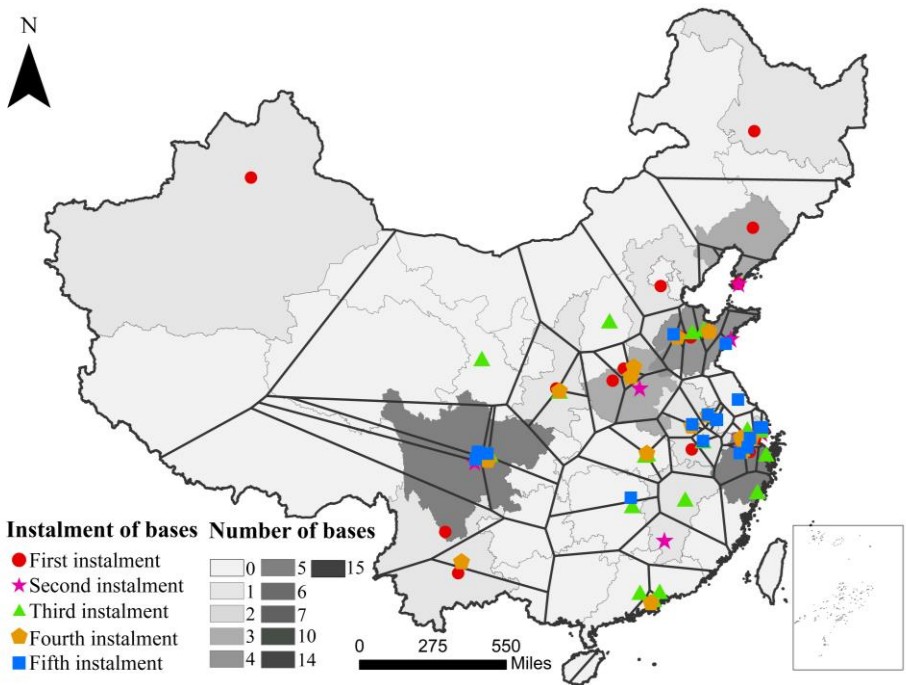

**Figure 2.** Spatial distribution of the smart healthy aging center.

In pursuit of a more profound analysis and elucidation of the spatial distribution modalities inherent in intelligent health and geriatric care facilities, we computed the coefficient of variation and the nearest neighbor index for these establishments (Table 1). The calculation results show that by 2021, the coefficient of variation of the smart health retirement base is $C_V$ = 281.33%, the nearest neighbor index is $R$ = 0.506, the Z score is −8.705, and the $p$ value is 0. Among them, $R$ < 1 and $C_V$ > 64% indicate that the spatial distribution type of the demonstration base is an agglomeration distribution; a Z score < −2.58 indicates that the possibility of randomly generating this clustering pattern is less than 1%, and the possibility of the demonstration base being an agglomeration distribution is greater than 99%.

From 2017 to 2021, the actual nearest neighbor distance of China's smart health care demonstration base was shortened by nearly 210 km.

The nearest neighbor index of the smart health care base in 2017 was $R = 0.928$, and the spatial structure showed a random distribution state. In 2018, 2019, 2020, and 2021, the nearest neighbour index values were $R = 0.839$, $R = 0.674$, $R = 0.589$, and $R = 0.506$, respectively, indicating that after 2017, the spatial taxonomy of China's intelligent healthcare hubs has initiated a trajectory towards clustering. This transition reflects a shift in the spatial distribution pattern from a stochastic arrangement to a discernible clustering disposition.

**Table 1.** The nearest neighbour index of the smart healthy aging center.

| Year | Number of Elderly Care Centers | Theoretical Nearest Neighbour Distance/km | Actual Nearest Neighbour Distance/km | Nearest Neighbour Index | Z-Score | Type of Space Structure |
|------|------|------|------|------|------|------|
| 2017 | 19 | 316.174 | 293.515 | 0.928 | −0.598 | Random |
| 2018 | 29 | 255.920 | 214.589 | 0.839 | −1.664 | Clustered |
| 2019 | 52 | 201.043 | 135.521 | 0.674 | −4.496 | Clustered |
| 2020 | 68 | 176.294 | 103.908 | 0.589 | −6.477 | Clustered |
| 2021 | 85 | 157.683 | 79.855 | 0.506 | −8.705 | Clustered |

*4.2. Aggregation Characteristics of Spatiotemporal Distribution*

Utilizing the geographical concentration index, we computed the indices of imbalance and geographical concentration pertaining to China's demonstration bases for intelligent healthcare and geriatric care throughout the period spanning 2017 to 2021. The calculation results show (Table 2) that at the provincial scale, the spatial distribution of China's smart health and elderly care bases is as follows: the imbalance and concentration are both significant. In the past five years, the number of bases in 44% of provinces has continued to be zero, and the remaining bases are mainly concentrated in Zhejiang, Sichuan, Shandong and other places. In terms of growth trends, the imbalance of demonstration bases will slow down in 2020 and 2021 compared with 2018 and 2019, and the overall trend will continue to rise. However, the concentration of demonstration bases has not increased significantly. The geographical concentration index $G$ has remained at around 30 from 2017 to 2021, indicating that the trend of concentration has not spread.

In addition, the spatial distribution imbalance ($S$) of the demonstration bases increases year by year, and the actual geographical concentration index ($G$) is always greater than the geographical concentration index ($G_0$) when distributed evenly, which also explains to a certain extent the spatial distribution of smart health care bases. Disequilibrium and agglomeration have the characteristics of sustained stability to a certain extent.

**Table 2.** The imbalance index and geographical concentration index of the smart healthy aging center.

| Year | Number of Elderly Care Centers | Number of Provincial Administrations | Imbalance Index (S) | Geographical Concentration Index (G) |
|------|------|------|------|------|
| 2017 | 19 | 12 | 0.263 | 32.015 |
| 2018 | 29 | 13 | 0.385 | 33.610 |
| 2019 | 52 | 19 | 0.442 | 30.285 |
| 2020 | 68 | 19 | 0.475 | 30.987 |
| 2021 | 85 | 19 | 0.492 | 31.282 |

*4.3. Density Characteristics of Spatiotemporal Distribution*

Leveraging the capabilities of ArcGIS 10.2 software, we undertook a sophisticated examination of the spatiotemporal distribution of smart healthcare and geriatric care facilities across China for the five-year period spanning from 2017 to 2021. Employing advanced statistical methodologies, including kernel density estimation and the standard deviation ellipse analysis, we unravelled the intricate patterns and trends within the dataset.

The fruit of this analytical labor is meticulously laid out in the sequence of graphical representations, as depicted in Figure 3a through Figure 3e.

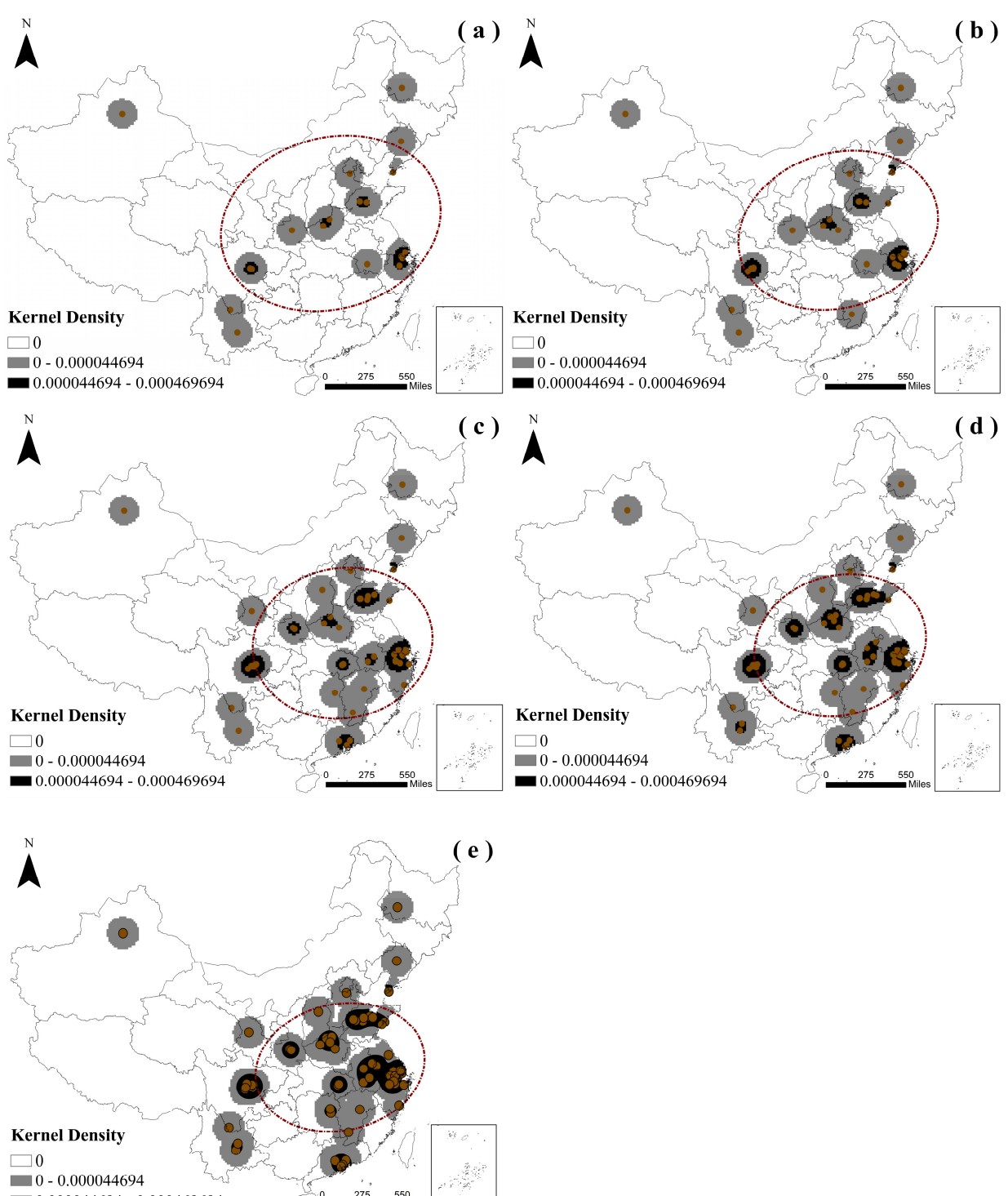

**Figure 3.** Kernel density and standard deviation elliptic of the smart healthy aging center: 2017–2021 corresponds to (**a**–**e**).

Through comparison, it was found that in 2017, there was no obvious density core in the study area. However, as the number and coverage of smart health and elderly care bases continue to increase, the spatial aggregation characteristics of the bases have become more and more obvious. In 2018 and 2019, the density growth of the three provinces of

Sichuan, Zhejiang, and Shandong changed significantly, and the base overall showed a three-pole concentration spatial pattern. In 2020 and 2021, while the three provinces of Sichuan, Zhejiang, and Shandong remain in the lead, the four provinces of Henan, Anhui, Guangdong, and Hubei have also begun to show agglomeration characteristics, and the base as a whole shows a three-pole concentration-multi-core surround type spatial pattern.

*4.4. Spatial Correlation Features*

Geo Data software was used to create a Moran scatter plot of China's smart health and elderly care bases from 2017 to 2021 (Figure 4), which can identify the spatial correlation patterns of each province.

The spatial autocorrelation as measured by Moran's I index experienced a remarkable surge, escalating from 0.011 in 2017 to 0.116, a nearly ten-fold increase. This substantial augmentation underscores the pronounced spatial interdependence characterizing the comprehensive dispersion of China's Smart Healthy Elderly Application Demonstration Bases.

Taking the most representative year of 2021 as an example, the first quadrant of its scatter plot is the six provinces of Zhejiang, Shanghai, Shandong, Henan, Guangdong, and Hubei. While these provinces have high observation values themselves, they are also adjacent to other provinces with high observation values. They have a positive spatial spillover effect and can provide a positive impact on the future development of the study area. Shaanxi, Liaoning, Jiangsu, Anhui, and Sichuan are located in the fourth quadrant of the scatter plot, which shows that these provinces themselves have relatively good development levels, but the surrounding areas have poor development. Hunan, Jiangxi, Xinjiang, Shanxi, Beijing, Tianjin, Inner Mongolia, and Fujian are located in the second quadrant of the scatter diagram. They have a low level of development and good development in surrounding areas. Chongqing, Guizhou, Jilin, Ningxia, Qinghai, Tibet, Gansu, Heilongjiang, Yunnan, and Hebei are located in the third quadrant of the scatter diagram, indicating that the development level of these provinces and their surrounding areas is low.

Conclusively, akin to the spatial layout of the elderly demographic, China's Smart Healthy Elderly Application Demonstration Bases conspicuously manifest an affluence concentration in the east and a scarcity disposition in the west [32]. The preponderance of the six most developed provinces primarily inhabits the eastern domain (Zhejiang, Shanghai, Shandong, Henan, Guangdong, and Hubei), all ranking within the upper echelons of China's GDP hierarchy. In contrast, the majority of the ten least developed provinces predominantly align with the western expanse (Chongqing, Guizhou, Jilin, Ningxia, Qinghai, Tibet, Gansu, Heilongjiang, Yunnan, and Hebei), with 70% of these provinces situated among the bottommost tier of China's GDP ladder [33]. This underscores the palpable impact of regional economic advancement on the establishment and progress of geriatric care amenities, along with a discernible affirmative correlation between economic progression and the evolution of sophisticated eldercare facilities [34]. Moreover, the eastern realm, marked by a dense population and hospitable natural environs, engenders a pressing demand for elderly care. Significantly, the eastern region's economic prosperity and resource abundance surpass those of its western counterpart [35], endowing it with greater fiscal solvency and a more vigorous resolve and capacity to furnish eldercare services, thereby fostering superior eldercare infrastructure development in the east.

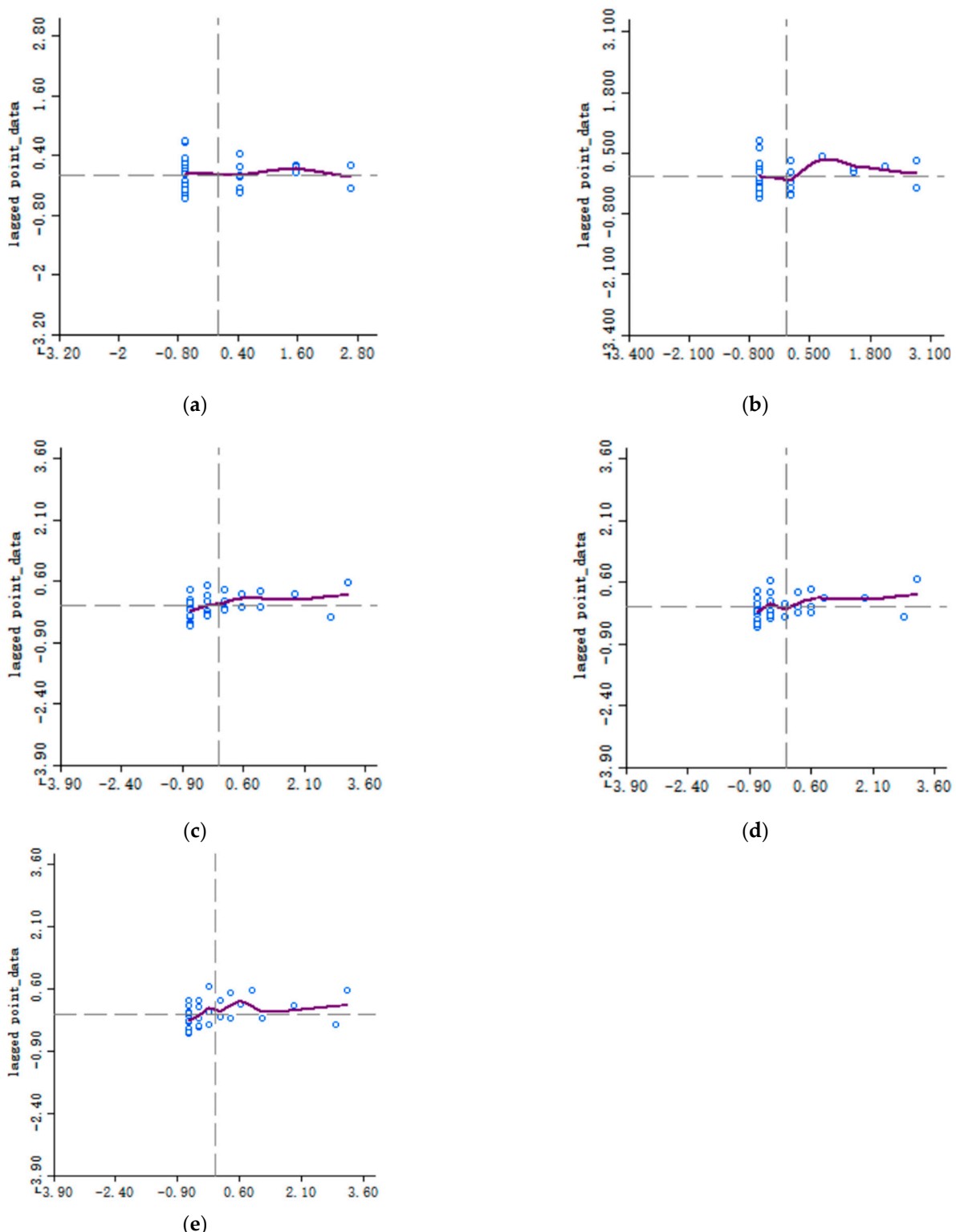

**Figure 4.** Moran scatter plot of the smart healthy aging center: 2017–2021 corresponds to (**a**–**e**).

## 5. Discussion of Sustainable Development Mechanism

The sustainable development mechanism governing smart health and elderly care demonstration bases encompasses a series of institutional, policy, technological, and managerial measures meticulously constructed to foster the sustainable advancement of elderly care demonstration bases within the framework of integrating smart technology and elderly care services. The fundamental objective of this mechanism is to guarantee that the elderly

care base delivers superior services while preserving long-term sustainability and elevating its developmental prowess without compromising future resources or the environment (Figure 5). This comprehensive study primarily concentrates on propelling the establishment and sustainable evolution of the sustainable development mechanism of smart health and elderly care demonstration based on three distinctive facets: the market-driven development mechanism, policy-driven development mechanism, and technology-driven development mechanism, all of which hold paramount significance in enhancing the quality of life for the elderly and fostering social harmony and stability. By adeptly harmonizing the interplay between the economy, society, and the environment, the sustainable development mechanism plays a pivotal role in constructing a comprehensive, salubrious, and intelligent elderly care service system.

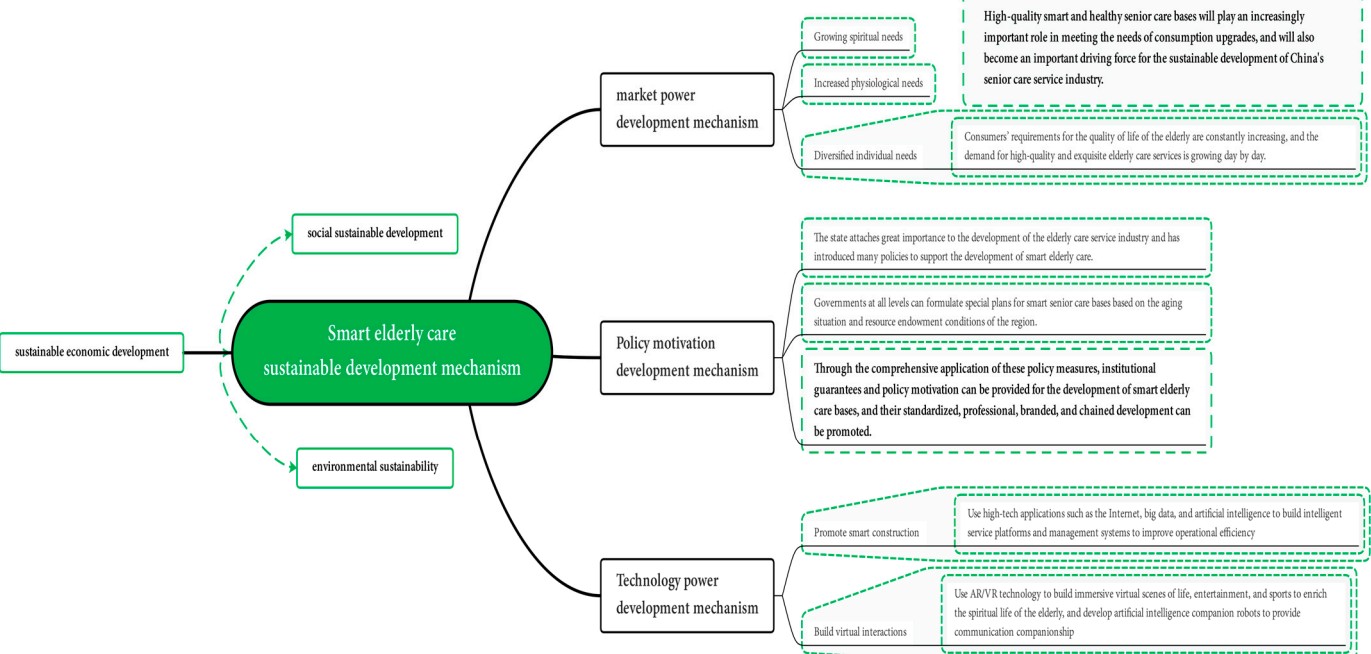

**Figure 5.** Description of the results of the sustainable development mechanism.

*5.1. Market Power Development Mechanism*

As China transitions into an ageing society, the demographic of individuals aged 65 and above continues to expand, concomitant with the steadily rising income levels of residents. Consequently, there has been a burgeoning demand from consumers for an elevated quality of life for the elderly populace, culminating in the emergence of well-crafted smart health and elderly care bases. These elderly care bases embody scientifically tailored layouts, well-endowed facilities, and personalized services that authentically cater to the spiritual and physiological needs of the elderly [36].

Smart health and elderly care demonstration bases are equipped to deliver remote medical services, routine physical examinations, and health monitoring to safeguard the fundamental physiological well-being of the elderly. For instance, through a remote medical platform, the elderly can seek medical advice and consult doctors without necessitating an in-person hospital visit. Addressing the physiological requirements, rehabilitation services assume the utmost importance. Smart health and elderly care bases harness technologies such as virtual reality (VR) and intelligent rehabilitation equipment to furnish bespoke rehabilitation programs. Furthermore, addressing mental health needs necessitates the provision of psychological support. Smart health and elderly care bases can seamlessly integrate psychological health applications and remote psychological counselling services. For instance, the elderly population can engage in meditation and relaxation exercises via

intelligent applications or arrange professional psychological consultations anytime and anywhere to alleviate psychological stress.

Elderly care bases of exceptional quality will institute scientifically standardized operational management systems, realizing community-based, information-based, and professional operational models, providing the elderly with personalized and compassionate services, and enabling them to lead wholesome and contented lives. Personnel will curate diverse product combinations tailored to the distinct physical conditions and interests of various elderly individuals, thereby delivering differentiated services. Embracing flexible business models such as membership and on-demand services broadens entry conditions and expands coverage [37].

Concurrently, the incessant enhancement of service processes and quality nurtures a distinctive brand image to augment brand appeal. It is foreseeable that high-caliber smart health and elderly care bases will progressively assume a pivotal role in fulfilling the demands of enhanced consumption, emerging as a substantial impetus for the sustained development of China's elderly care service industry.

*5.2. Policy Motivation Development Mechanism*

The nation accords utmost priority to the progression of the elderly care service industry and has promulgated an array of policies to bolster the advancement of smart elderly care, encompassing financial subsidies, tax incentives, and priority land allocation, thereby laying a robust policy foundation for the market expansion of smart health and elderly care bases [38–40].

Administrative bodies at all echelons are empowered to devise specialized plans for smart elderly care bases contingent upon the local ageing demographics and resource endowments, delineating development concepts, spatial layout, funding arrangements, etc., thus furnishing top-tier design and guiding ideology [41].

Augmenting regulations and standards assumes paramount importance. Presently, China's elderly care policies and regulations predominantly revolve around safeguarding the legal rights and interests of the elderly, exemplified by the "Law on the Protection of the Rights and Interests of the Elderly", a specific piece of legislation in China designed to safeguard the rights and interests of the elderly [42]. This law stipulates efforts to fortify the protection of the rights and interests of the elderly, the provision of convenient services, and the lawful protection of their legitimate rights and interests. Regarding smart elderly care, this law mandates government entities to fortify information services for the elderly, incentivize societal forces to undertake information services for the elderly, enhance the information literacy of the elderly, and ensure the information security of the elderly. The "National Plan for the Protection of the Rights and Interests of the Elderly (2017–2022)", jointly issued by nine departments, including the National Health Commission, the Ministry of Civil Affairs, and the National Development and Reform Commission, explicitly underscores the imperative to promote information services for the elderly, elevate the information literacy and information security capabilities of the elderly, and establish an information service system for the elderly [43]. These regulations and standards provide a robust legal foundation for the development of smart health and elderly care demonstration bases, steering the wholesome and methodical growth of this industry.

The pertinent legislation and regulations governing intelligent elderly care, albeit somewhat bolstering the establishment of digitalization in this field, currently suffer from structural inadequacies, lacking a comprehensive institutional framework. Consequently, it is imperative to institute and refine, from national to local levels, a supervisory system for intelligent elderly care facilities, culminating in the formulation of a standardized regulatory framework to ensure their uniform construction and operation. This undertaking encompasses elucidating the rights and responsibilities of the government, operators, and elderly residents. Dedicated funds should be allocated for the establishment of intelligent elderly care facilities, furnishing financial backing through service procurement, subsidies, and interest-based incentives. Facilitating social capital involvement is equally paramount.

Furthermore, endorsing the adoption of public–private partnership and government–social capital cooperation models is essential to engender a multifaceted investment and supply operational paradigm. These initiatives furnish robust legal safeguards for the advancement of intelligent elderly care, enhancing the well-being and security of the elderly populace.

The enhancement of incentive mechanisms is imperative. Commendations and rewards ought to be bestowed upon intelligent elderly care facilities exhibiting commendable operational efficacy and superior service quality. The creation of a policy guidance compendium is instrumental in curbing non-standard operational practices. Additionally, fostering a robust workforce specializing in elderly care services, coupled with the implementation of training and evaluation frameworks, is crucial in providing human resource support for the development of intelligent elderly care facilities.

By judiciously implementing these policy measures, one can furnish institutional assurances and policy momentum for the growth of intelligent elderly care facilities, thus advocating their standardized, professional, branded, and interconnected evolution.

*5.3. Technology Power Development Mechanism*

The swift evolution of information technology has bestowed unparalleled opportunities for smart health and elderly care. Innovations such as the Internet of Things (IoT), artificial intelligence (AI), big data, cloud computing, and 5G have found extensive applications in elderly care services, catalyzing the metamorphosis and enhancement of traditional elderly care services into intelligent elderly care services.

The IoT technology is harnessed to achieve intelligent monitoring and management of the daily lives of the elderly. Through the utilization of wearable devices and sensor networks, real-time collection and analysis of the health data of elderly individuals can be achieved. For instance, smart wearable devices can monitor heart rate, blood pressure, sleep quality, and physical activity, providing invaluable data for health assessment and early detection of potential health risks.

AI technology is leveraged to deliver personalized care and assistance. AI-powered virtual assistants can engage with the elderly, prompt them regarding medication schedules, offer entertainment, and even extend companionship to assuage loneliness. Big data and cloud computing are deployed to institute comprehensive databases and platforms for the storage and analysis of elderly health records, fostering the sharing of medical resources and optimizing service allocation. The advent of 5G technology engenders high-speed, low-latency communication, facilitating real-time remote consultations, telemedicine, and interactive experiences, markedly enhancing the efficiency and caliber of healthcare services for the elderly [44].

In summation, the sustainable development mechanism of smart health and elderly care demonstration bases embodies a holistic approach that amalgamates market forces, governmental support, and technological innovation to guarantee the enduring viability and efficacy of elderly care services [45]. This mechanism serves as a template for propelling the sustainable progression of the elderly care industry in the epoch of smart technology, contributing to the well-being and quality of life of the elderly populace [46].

## 6. Discussion and Conclusions

*6.1. Discussion*

This study aims to deeply explore the spatial and temporal distribution evolution and sustainable development mechanisms of the GIS-based smart healthy ageing demonstration base. Through GIS technology, we track the changes in the spatial and temporal dimensions of the Smart Healthy Aging Demonstration Base and deeply analyze the rationality of its spatial layout, the evolution of its geographic characteristics, and its impact on the surrounding environment. We aimed to explore the sustainable development mechanism of the Smart Healthy Aging Demonstration Base in terms of economic, social, and environmental aspects to establish a comprehensive theoretical framework that will lead

the base to balance economic benefits, social responsibility, and environmental protection in the long term.

The closest neighbour index method was used to analyze the agglomeration of the bases; the spatial extent was analyzed by Tyson polygons; the hotspot distribution of the bases was examined by applying kernel density; and the aggregation characteristics of the distribution of the bases were examined by spatial autocorrelation analysis. A quantitative analysis based on the aforementioned research framework yields detailed spatial analysis results, effectively quantifying the characteristics of regional spatial relationships, such as the balance, disparities, and patterns of agglomeration within the distribution of smart healthy ageing demonstration bases across 31 Chinese provinces. The visual representation of graphical data enhances the intuitiveness and readability of the research findings. Utilizing a combination of diverse analytical methods, this study showcases the spatiotemporal distribution characteristics of China's intelligent healthy ageing demonstration bases from varied perspectives, thereby facilitating a comprehensive and precise assessment of the spatial and temporal distribution of the ageing demonstration bases and augmenting the scientific and persuasive nature of the study. It offers novel perspectives for appraising the development of China's intelligent, healthy ageing demonstration bases and furnishes scientific argumentation to uphold the research findings.

It was found that China's intelligent healthy ageing bases are spatially clustered, and the clustering areas are mainly located in the central and eastern parts of China, with Zhejiang, Sichuan, and Shandong provinces and regions being the most significant. The bases as a whole show a spatial pattern of the "three-pole concentration-multi-core encircling type". The spatial correlation Moran I index increased from 0.011 in 2017 to 0.116, a nearly 10-fold increase, indicating that China's intelligent, healthy ageing bases also have a certain degree of spatial correlation on the whole. Finally, the study proposes that the market power development mechanism, the policy power development mechanism, the technology power development mechanism, and the other "three major power" mechanisms correspond to each other and continue to promote the construction of wisdom to realize the sustainable development of the demonstration base. This study provides a scientific basis for the planning layout and industrial development of relevant departments.

Compared with previous studies, this study analyzes the relationship between the base and the surrounding environment in more detail using GIS technology and digs deeper into the potential impact of spatial layout on the service effect. It pays more attention to the analysis of sustainability, emphasizing the important significance of comprehensively promoting the establishment and sustainable development of the sustainable development mechanism of the demonstration base for smart healthy ageing from the three aspects of the market-powered development mechanism, the policy-powered development mechanism, and the technological-powered development mechanism, improving the quality of life of the elderly and promoting the harmony and stability of society. The Smart Healthy Aging Application Demonstration Base will establish a sound mechanism for dynamic monitoring of the health of the elderly through the integration of high-quality medical resources and social ageing service resources, promote the in-depth application of information technology and intelligent hardware in the ageing service of the elderly, and enhance the sense of accessibility and well-being of the elderly, which is of great significance for the establishment of a long-term, high-quality ageing support service system.

*6.2. Conclusions*

(1) China's intelligent, healthy ageing bases are spatially clustered, and the clustered areas are mainly located in the central and eastern parts of China, with the overall spatial distribution proving to be dense in the east and sparse in the west, which is highly similar to the spatial distribution pattern of the elderly population. Zhejiang, Sichuan, and Shandong provinces and regions are the most significant. The base as a whole shows a spatial pattern of the "triple-pole concentration-multi-core encircling type".

(2) From 2017 to 2021, the proximal distance of China's intelligent healthcare demonstration sites has been reduced by almost 210 kilometres. In 2017, the nearest neighbour index ($R$) of the intelligent healthcare base was 0.928, signifying a spatial structure in a state of random distribution. The nearest neighbour indices for 2018, 2019, 2020, and 2021 were 0.839, 0.674, 0.589, and 0.506, sequentially revealing that post-2017, the spatial nature of China's intelligent healthcare bases commenced continued manifestation as an agglomeration type, with the spatial distribution pattern transitioning from a random dispersion to an agglomeration dispersion.

(3) The spatial correlation The Moran I index increased from 0.011 in 2017 to 0.116, an increase of nearly 10 times, indicating that China's smart healthcare bases also have a certain degree of spatial correlation as a whole.

*6.3. Shortcomings and Prospects*

The GIS-based smart healthy ageing demonstration base is a hot spot of current social concern because it combines technology and healthy ageing to provide a better quality of life for the elderly. However, despite the progress made in this study, there are still shortcomings in the research. The spatial and temporal distribution and sustainable development mechanisms of smart healthy ageing demonstration bases need a large amount of data support, but the completeness and quality of the current data are still problematic, limiting the accuracy and feasibility of decision-making. The connotation of smart ageing development is not explored deeply and comprehensively enough in the study. The construction of a demonstration base for smart, healthy ageing needs to be further improved. The future policy recommendations for smart ageing that can be concretized and operable are not strong.

When analyzing the sustainable development mechanism, the market power development mechanism, the policy power development mechanism, and the technology power development mechanism from three aspects comprehensively promote the establishment of the sustainable development mechanism and sustainable development of the intelligent healthy ageing demonstration base.

However, the potential limitation is the complexity of environmental variables. There are numerous environmental factors affecting the sustainable development of the base, including climate change, geological conditions, etc., and the interplay of these factors may be beyond the scope of the study, leading to difficulties in grasping environmental sustainability in a comprehensive manner. Another potential limitation is the sample bias. The sample points are 89 smart healthy ageing demonstration bases, and the number of new demonstration base points will continue to be announced, which may not be sufficient to represent the diversity of the industry as a whole. The generalization of the study's findings may be limited by sample selection due to regional differences and the diversity of senior care bases. Another limitation is that future uncertainties, such as policy changes and economic fluctuations, which may have a significant impact on the development of smart senior living bases, were not adequately addressed, and the future uncertainties may affect the feasibility and usefulness of the study.

When collecting and analyzing the spatial and temporal distribution data of GIS-based smart healthy ageing demonstration bases, the privacy of individuals or organizations may be involved. Therefore, this study fully considers and follows the relevant privacy regulations and ethical guidelines. In future studies, the key factor of protecting data sensitivity and ensuring that the research is conducted within a responsible and ethical framework will be fully considered.

Regarding the existence of social justice issues that should be taken into account in the study, the analysis of GIS may reveal differences between different regions, which may involve issues such as socio-economic status, resource allocation, and so on. Therefore, unjust qualitative or quantitative evaluations of certain communities or groups should be avoided in future research to avoid reinforcing social inequalities. The potential impact of these ethical considerations is a key factor in ensuring that research is conducted within a

responsible and ethical framework. Therefore, ethical principles should always be placed at the core of research practice to ensure the sustainability and social value of research.

In the future, in subsequent eras, the completeness of data may be augmented by the establishment of additional data collection facilities and data-sharing mechanisms with the aim of bolstering more precise decision-making and advancement. The propagation of intelligent, salubrious ageing technologies across diverse regions could be facilitated through instruction and technical support to attain a harmonious evolution in elderly care services. Fostering collaboration among various sectors, spanning from government bodies and healthcare organizations to research institutes and technology enterprises, is essential for collectively propelling sustainable progress at the Smart Healthy Aging Demonstration Base.

Augmenting the assimilation of intelligent technologies, encompassing artificial intelligence, the Internet of Things, and extensive data analysis, is pivotal for elevating the operational standard of smart healthy ageing demonstration centers. Research endeavors might pivot towards health monitoring systems and adaptive environmental technologies to cultivate a secure, convenient, and astute environment for the aged populace.

Given the multifaceted ageing requisites of the elderly, forthcoming investigations could delve deeper into the influence of socio-cultural elements on intelligent, healthful ageing. This encompasses an exploration of service necessities, communal support structures, and the acceptance of technology among older individuals within distinct cultural milieus. These factors are poised to contribute to the formulation of culturally tailored models for elderly care services. Subsequent research initiatives could intensify the scrutiny of policies and regulations related to intelligent and healthful ageing demonstration sites. A keen understanding of the impact of policy frameworks on the elder care industry, including incentives, regulatory mechanisms, and legal accountabilities, is imperative for formulating targeted policy recommendations for intelligent elderly care.

Furthermore, there is a need for refining methodologies and benchmarks for evaluating the sustainability of intelligent, healthful ageing demonstration centers. This might encompass the development of comprehensive assessment tools covering economic, social, and environmental dimensions, along with flexible evaluation frameworks adaptable to diverse regions and specific ageing scenarios.

Future research should underscore interdisciplinary cooperation, amalgamating expertise across domains such as geography, medicine, sociology, and information technology. This approach will engender a comprehensive and profound comprehension of the multifaceted facets of intelligent ageing, fostering the exchange and amalgamation of knowledge from varied disciplines. Exploring these potential research domains will propel the domain of intelligent, healthful ageing, advancing service quality, catering to diverse elderly necessities, and spurring innovation in sustainable ageing services worldwide.

Conclusively, GIS-based intelligent, healthful ageing demonstration sites are evolving; however, several impediments persist that necessitate surmounting to realize enhanced ageing services and sustainable development. Prospective efforts should concentrate on data accuracy, technology diffusion, privacy safeguards, and multi-disciplinary collaboration.

**Author Contributions:** Methodology, X.C., B.C., C.U.I.W. and H.Z.; software, X.C. and B.C.; writing—original draft, X.C. and B.C.; writing—review and editing, X.C. and B.C. All authors have read and agreed to the published version of the manuscript.

**Funding:** This research received no external funding.

**Institutional Review Board Statement:** Not applicable.

**Informed Consent Statement:** Not applicable.

**Data Availability Statement:** The data presented in this study are available on request from the corresponding author. The data are not publicly available due to the Chinese government protects the privacy of the elderly care industry and geographic data.

**Acknowledgments:** This research acknowledges the support of Macao Polytechnic University (RP/FCHS-01/2023).

**Conflicts of Interest:** The authors declare no conflict of interest.

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
