# Peer review of "Research on the Spatial and Temporal Distribution Evolution and Sustainable Development Mechanism of Smart Health and Elderly Care Demonstration Bases Based on GIS"

_applsci, doi:10.3390/app14020780_

Round 1

Reviewer 1 Report

Comments and Suggestions for Authors

1. The basic uses and calculation methods of the research method are introduced in this paper. It is suggested to further explain why these research methods are used in this paper. To reflect the scientific selection of research methods.

2. In the fifth part of this paper, the sustainable development mechanism is discussed, but the relationship between the thesis and the research results of this paper is vague. It is suggested to clarify the relationship between the proposed argument and the results of this paper more clearly in the discussion, so as to reflect the rigor of the writing logic.

3. To improve the readability of the paper, it is suggested to draw a graph of the conclusions of the sustainable development mechanism proposed in this paper.

Comments on the Quality of English Language

English language and style are fine/minor spell check required

Reviewer 2 Report

Comments and Suggestions for Authors

The article entitled Research on the spatial and temporal distribution evolution and sustainable development mechanism of smart health and elderly care demonstration bases based on GIS is interesting first of all because it addresses a topic of high present interest starting from a serious issue which has affected China for some time: demographic aging.

The article contains the appropriate structure. It is correctly divided into relevant sections and their content coincides with their titles. Bibliography is correctly formulated. English style is generally good, with only minor flaws. The language of the article is mature, but it sometimes needs adjusting. The sentences comprised between lines 364 and 391 have no subjects, they are just an enumeration of predicates mainly used imperatively, lacking subjects. The same happens in other paragraphs (as for example, in lines 89, 144-145, 252-253), which means that the whole text must be read again and modified.

The abstract should better clarify the objectives and results of the study. I rather see the methods presented extensively, but not the aim and finality of the paper.

In the Introduction section it is advisable to include more statistics on the Chinese population aging in order to better grasp the utility of such a topic.

All figures should be a little bit adjusted by increasing the font of the legend which is unreadable in their present shape.

You must also pay attention to tables. The phrase  1 Tables may have a footer” should be removed from below tables. Table 2 does not have a title.

Line 311 – can you support your statement by including some statistics of income levels?

Can you find some (economic, social etc) factors that explain the distributions illustrated by the figures?

Line 331 – please, expand a little the policies adopted to support the development of smart elderly care.

Line 340 – What is the present legal and regulatory framework in China?

I would not replace the Conclusions section with Shortcomings and Prospects. Conclusions are very important because they show us the novelty of the paper, its added value.

Comments on the Quality of English Language

English style is generally good, but the whole text must be revised because some sentences lack subjects.

Reviewer 3 Report

Comments and Suggestions for Authors

In this article, the authors explored the spatial and temporal distribution evolution and sustainable development mechanism of smart health and elderly care demonstration bases in China.

The article lacks a clear and explicit statement regarding the potential limitations of the research findings, which is essential for providing a comprehensive understanding of the study's scope and applicability.

 Additionally, the article does not address potential biases or confounding variables that may have influenced the results, which is crucial for ensuring the validity and reliability of the research.

Furthermore, the article does not discuss any ethical considerations or potential implications of the research findings, which is important for promoting responsible and ethical research practices.

The article would benefit from a more detailed discussion of the potential limitations of the research findings. This could include addressing any potential biases or confounding variables that may have influenced the results, as well as ethical considerations and implications of the research findings.

The article could provide a more comprehensive overview of the research area, including a clear statement of the research scope and applicability.

Finally, the article could benefit from a more thorough discussion of the prospects for future research in this area, including potential areas for further investigation and development.

Round 2

Reviewer 3 Report

Comments and Suggestions for Authors

After going through the responses of the authors, I am is of the view that the authors have incorporated the comments of the undersigned. I think, no more changes are required in the article.